# Recent Advances in Nanovaccines Using Biomimetic Immunomodulatory Materials

**DOI:** 10.3390/pharmaceutics11100534

**Published:** 2019-10-14

**Authors:** Veena Vijayan, Adityanarayan Mohapatra, Saji Uthaman, In-Kyu Park

**Affiliations:** 1Department of Biomedical Sciences, Chonnam National University Medical School, Gwangju 58128, Korea; veenavj4392@gmail.com (V.V.); mkaditya55@gmail.com (A.M.); 2Department of Polymer Science and Engineering, Chungnam National University, 99 Daehak-ro, Yuseong-gu, Daejeon 34134, Korea; sajiuthaman@gmail.com

**Keywords:** nanovaccines, biomimetic, antigens, adjuvants, antigen-presenting cells

## Abstract

The development of vaccines plays a vital role in the effective control of several fatal diseases. However, effective prophylactic and therapeutic vaccines have yet to be developed for completely curing deadly diseases, such as cancer, malaria, HIV, and serious microbial infections. Thus, suitable vaccine candidates need to be designed to elicit appropriate immune responses. Nanotechnology has been found to play a unique role in the design of vaccines, providing them with enhanced specificity and potency. Nano-scaled materials, such as virus-like particles, liposomes, polymeric nanoparticles (NPs), and protein NPs, have received considerable attention over the past decade as potential carriers for the delivery of vaccine antigens and adjuvants, due to their beneficial advantages, like improved antigen stability, targeted delivery, and long-time release, for which antigens/adjuvants are either encapsulated within, or decorated on, the NP surface. Flexibility in the design of nanomedicine allows for the programming of immune responses, thereby addressing the many challenges encountered in vaccine development. Biomimetic NPs have emerged as innovative natural mimicking biosystems that can be used for a wide range of biomedical applications. In this review, we discuss the recent advances in biomimetic nanovaccines, and their use in anti-bacterial therapy, anti-HIV therapy, anti-malarial therapy, anti-melittin therapy, and anti-tumor immunity.

## 1. Introduction

Our immune system comprises a complex network of cells, tissues, and organs that work in harmony to protect the body against deadly diseases. This immune system attacks and eliminates foreign invading particles with exquisite specificity. Diseases are caused by malfunctioning or underperforming immune response; an over-reactive immune system can cause autoimmunity, which may lead to the destruction of healthy tissue [1,2], and an underactive immune system can make our body more susceptible to infection [3]. Vaccines consist of a biological agent that resembles a disease-causing microorganism and improves immunity against that particular disease. They develop immunity that can control and adjust unbalanced immune systems that are either overreactive or underactive [4,5,6]. Weiner et al. described the first therapeutic vaccine against autoimmunity [7]. The development of vaccines has historically been based on Louis Pasteur’s “isolate, inactivate, inject” paradigm [8]. Currently, vaccines are considered to be one of the most effective tools for the prevention of infectious diseases. Thus, vaccine developments against bacterial infections, viral infections, and cancer are considered to be significant milestones in the field of medicine [9]. In the past, traditional vaccines made from pathogens in either killed or inactivated forms were considered efficient [8,10]. Vaccines with live-attenuated pathogens make use of the weakened type of microbe to create a stronger and enduring immune response. However, a significant concern arises when the weak pathogen might revert to its active form, causing severe disease condition. The mutagenic actions inside the infected host organism could generate more virulent strains. In addition, pathogens in their inactive or killed forms cannot revert to active forms, and tend to stimulate weaker immune responses, which in turn requires multiple administrations of doses, hence limiting its practical use.

Usually, vaccines are comprised of an antigen, which acts as the target for causing immune response, and an adjuvant, which is co-administered with the antigen to enhance the immune response. Aluminum was the first-ever adjuvant, and was used primarily to increase antibody production, making it a suitable candidate for vaccine formulation [11]. However, aluminum adjuvants fail to generate strong cell-mediated immunity, and carry the risk of autoimmunity, and long-term brain inflammation, causing adverse health issues. Adjuvants have been reported to cause both local as well as systemic toxicity. Certain adjuvants like Freund’s incomplete adjuvant, Quil A, induce local toxicity, whereas adjuvants based on pathogen associated molecular patterns like Aluminum adjuvants induce systemic toxicity [12]. Another adjuvant was Freund’s incomplete adjuvant in the form of mineral oil-in-water emulsion that contained heat-killed mycobacteria; it was found to be reactogenic in humans [13]. Although these adjuvants could induce local immune reactions, they failed to generate strong cell-mediated immunity, which demanded the development of new adjuvants for successful vaccine delivery. According to a web-based central database pertaining to vaccine adjuvants, nearly a hundred vaccine adjuvants have been used in various vaccines against different pathogens, of which very few have received licenses for human use [14].

Recently, nanoparticles (NPs) have gained enormous attention as delivery vehicles for vaccines. Nanovaccine formulations not only provide enhanced antigen stability and immunogenicity, but also offer targeted delivery and prolonged release. A high number of NP vaccines with varied physicochemical characteristics and properties have been approved for clinical use [15,16,17]. The primary purpose of the use of nano- and microparticle-based delivery systems is to enhance the duration of antigen presentation and dendritic cell (DC)-mediated antigen uptake, which result in the direct stimulation of DCs, and promote cross-presentation [15,16,18]. Furthermore, NPs help in protecting the antigen and adjuvant from premature enzymatic and proteolytic degradation [19]. Vaccine antigens can be delivered to the target site by either encapsulating them inside an NP, or by decorating them onto the surface of NPs. The NP delivery systems can load multiple components in a single carrier, which enable a prolonged, simultaneous, and targeted delivery of antigens [20,21], adjuvants [22,23], DNA plasmids [24], and detained bacterial toxins [25]. The development of vaccine candidates is based on several factors, such as minimalist compositions, low immunogenicity, and formulations that boost antigen effectiveness [26,27,28,29]. Owing to their unique physicochemical characteristics, such as large surface area-to-volume ratio, controllable size and shape with different surface charge, NPs can be surface-engineered with peptides, proteins, polymers, cell-penetrating peptides, and other targeting ligands, which make them a versatile delivery vehicle for vaccine formulations. Design of NPs based vaccines can assist for multimodal imaging to improve therapeutic level by visualizing the vaccine inside our body [30,31,32,33]. Although NPs have the abovementioned advantages, they have disadvantages, in that they lack colloidal stability in physiological conditions due to protein corona formations, and have undesirable interaction with the reticuloendothelial system (RES) [34,35].

Biomimetic NPs are a novel class of NPs that exhibit enhanced colloidal stability, while efficiently avoiding unwanted interaction with immune cells like RES, and prolonging circulation in the blood [36,37,38]. These nanovaccines involve carrier NPs that mimic biological membranes, and when administered in the body, achieve prolonged circulation and evasion of immune responses [39]. Among biomimetic NPs, liposomes are obtained by the dispersion of phospholipids in water, and have a high loading capacity, with the ability to co-deliver both hydrophobic and hydrophilic drugs [40]. Cell-membrane coated NPs are another type of biomimetic nanocarrier with a “core–shell” structure, in which the NP forms the hydrophobic core, and a thin layer of plasma membrane coating acts as the shell. In this NP, various cell membranes are used to cloak synthetic NPs through a top-down fabrication method, thus preserving the physicochemical properties of the core synthetic NPs, while maintaining the cellular composition on its hydrophilic membrane shell [41]. Hu et al. [42] reported the first membrane-coated NPs where red blood cell (RBC) membranes were coated over a polymeric NP through extrusion [42]. Many types of membranes from different sources, such as RBCs [41,42,43], leukocytes [44,45,46], cytotoxic T-cells [47], NK cells [48], platelets [49], macrophages [44,50], and cancer cells [51,52], have been used in the preparation of membrane-coated NPs. Another type of biomimetic nanovaccine can be self-assembling proteins, which are known to have high symmetry and stability, and can be structurally organized into particles of sizes (10–150) nm [53,54]. These self-assembling protein NPs play diverse physiological roles, and are selected as vaccine carriers owing to their ability to self-assemble and deploy into a definite structure that mimics a natural microbe architecture [55]. Virus-like particles (VLPs) are another type of biomimetic nanovaccine that contain noninfectious subsets of virus that lack genetic materials; they assemble without containing any viral RNA [56].

In this review, we discuss the recent advances in biomimetic nanovaccines and their applications in anti-bacterial therapy, anti-HIV therapy, anti-malarial therapy, anti-melittin therapy, and anti-tumor immunity.

## 2. Components of Biomimetic Immunomodulatory Nanovaccines

Biomimetic nanovaccines include a biomimetic carrier that is loaded with therapeutic molecules that are designed to deliver to the target site. Figure 1 shows that the various types of biomimetic NPs involve liposomes, protein NPs, cell-membrane decorated NPs, and VLPs.

### 2.1. Types of Biomimetic Nanoparticles (Nps)

Table 1 describes the different type of biomimetic nanovaccines that are reported so far along with its applications.

#### 2.1.1. Liposomes

Liposomes are biomimetic products that are formed by dispersing phospholipids in water [58,62,63]. They occur as either unilamellar vesicles with a single phospholipid bilayer, or as multilamellar vesicles with several concentric phospholipid shells separated by different layers of water. Liposomes can be modified to incorporate both hydrophobic and hydrophilic molecules into the phospholipid bilayer and aqueous core [64]. Liposomes can be used to encapsulate antigens within their core for delivery. They form virosomes when viral envelope glycoproteins are incorporated into their base [65,66]. Influenza virus was the primary focus for virosome studies which has been established for industrial application as human vaccine [67]. Five vaccines based on virosome are under clinical trials, and four virosome vaccines are approved for commercial application in various diseases [67]. One of the commonly used NPs for adjuvant delivery in DNA vaccines is liposome-polycation-DNA NPs; they are formed by the combination of cationic liposomes and cationic polymer-condensed DNA. Liposome-polycation-DNA assembles to form a nanostructure, with the condensed DNA located inside the liposome with a size of 150 nm [24,68]. Moon et al. [21] reported the development of a malaria vaccine, which could be used for the delivery of polymeric PLGA NPs enveloped with lipid antigens. In their work, Moon and coworkers developed a pathogen-mimicking nanovaccine, in which the candidate malarial antigen was conjugated to the lipid membrane and incorporated with an immunostimulatory molecule, monophosphoryl lipid A-MPLA, and further used to elicit immune responses against *P. vivax sporozoites* [21]. Yang et al. [52] used cancer cell membranes which were modified with lipids using the lipid-anchoring method, and then further coated them over polymeric NPs with a toll-like receptor 7 (TLR 7). This biomimetic membrane nanocarrier was reported for use as an anticancer vaccine, as well as for the delivery of TLR 7 as an adjuvant [52].

#### 2.1.2. Virus-Like Particles (VLPs)

VLPs are molecules that resemble the structure of viruses without viral genetic material. These self-assembling NPs that lack infectious nucleic acid are formed by the self-assembly of biocompatible capsid proteins. They are ideal nanovaccine systems, as they have the innate viral structure, which can interact with the immune system without any threat of causing infections [69,70]. These VLPs can act as vaccines have nano-size and a repetitive structural order, and could induce an immune response in the absence of an adjuvant [71]. VLPs assemble without encapsulating any viral RNA, and hence they are noninfectious and nonreplicating, as the genes coded for viral integrase are deleted before expression. This prevents packed genome integration into the host cell, as well as the recombination of the live or defective virus. The first VLP vaccine was developed against the hepatitis B virus, which was later commercialized in 1986 [72]. VLP vaccines against hepatitis E and the human papillomavirus have been used in human since 2006 [73,74].

VLPs can be obtained from a variety of viruses, and can have different sizes ranging (20 to 800) nm; further, they can be obtained via different processes [56]. The initial approach to obtain VLPs involves the self-assembly of capsid proteins in the expression host, followed by purification of the assembled protein to avoid contaminants that are adhered or encapsulated. However, in a few cases, for better quality and low contamination, the VLP structure needs to be disassembled and reassembled. Another emerging method to obtain VLPs is to use cell-free in vitro processing, wherein at first large-scale purification is performed to prevent contamination, and then assembly of VLP structures in vitro, to avoid their disassembly in a cell; commercialized VLPs are derived from a target virus by self-assembling its proteins.

For VLP to be used as a delivery vehicle, the target antigen from a virus different from the one used in the VLP is attached to the VLP surface; and this surface-modified VLP paves the way for its use in targeting various diseases. VLPs could be engineered to attach additional proteins on its surface, either through the fusion of proteins on the particle, or by expressing multiple antigens, which in turn protects against its source virus and other antigens present on its surface [75]. Polysaccharides and small organic molecules are non-protein antigens that can be chemically attached to the VLP surface to form bioconjugate particles [76]. The baculovirus expression system is mostly used to generate VLPs with an excellent safety profile, as baculovirus does not naturally infect human [77]. In another study, a safe and efficient VLP system based on avian retrovirus was designed such that the system was considered safe, as it could not replicate itself in human cells. This system was considered as safe because the VLP constitutes only Gag fusion protein; a single VLP could deliver about (2000–5000) copies of the Gag fusion protein into the transduced cell. In another study, VLPs were created for delivery with two different approaches: the intracellular distribution of Gag fusion proteins, or by modifying the surface of VLPs for receptor/ligand-mediated delivery (Figure 2) [57].

#### 2.1.3. Self-assembling Protein Nanoparticles (NPs)

Many naturally occurring proteins can self-assemble to form NPs with high symmetry and stability, and these NPs are structurally organized to form particles that range in size (10–150) nm [53,54]. These NPs with diverse physiological roles are selected as vaccine carriers, owing to their ability to self-assemble and deploy into a definite structure that mimics a natural microbe architecture [55].

Ferritin is a protein that protects cells from damage caused by Fenton reactions, in which iron catalyzes hydrogen peroxide, and converts it into highly toxic hydroxyl radical. Under oxidizing conditions, harmful reactive oxygen species are produced from free Fe (II), which can damage cellular machinery [78]. Ferritin has a hollow structure, and the ability to store iron within this hollow cavity; thus, it acts as a storage system for iron [79]. Ferritin can self-assemble into spherical nanostructures and be used to fuse with the influenza virus haemagglutinin (HA) genetically, and the recombined protein spontaneously assembles into a particle of octahedral symmetry. This reforms into eight trimeric HA spikes, and elicits a stronger immune response, compared to an inactivated trivalent influenza virus [80]. Another type of self-assembling protein is the major vault protein (MVP). Champion et al. [81] reported that 96 units of the MVP self-assemble to form a barrel-shaped vault NP of length 70 nm and width 40 nm.

Further, they mentioned that genetically fused antigens that have minimal interactions could be loaded onto vault NPs that had self-assembled through mixing with MVPs. In their work, they encapsulated an immunogenic protein termed the major outer membrane protein of *Chlamydia muridarum* into hollow vault nanocapsules. These hollow vault nanocapsules were modified to bind IgG for an enhanced immune response, to induce protective immunity at distant mucosal surfaces [81]. Wahome et al. reported another self-assembling protein NP, an adjuvant-free immunogen [82] obtained by the self-assembly of a monomeric chain into an ordered oligomeric form as an antigen-presenting system that could be suitable for vaccines. This self-assembling protein NP was formed by incorporating the membrane-proximal external region (MPER) of HIV-1 gp41, which is identified as a target for a wide range of neutralizing antibodies, in the N-terminal pentamer, to produce an α-helical state of the 4E10 epitope, without causing structural changes in 2F5 epitopes. These self-assembled NPs showed enhanced membrane-proximal region-specific titers, owing to the presence of a repetitive antigen display of MPER even without any adjuvant, thus resulting in the formation of an adjuvant-free immunogen as a potential HIV vaccine [82].

#### 2.1.4. Cell Membrane-Decorated Nanoparticles (NPs)

As discussed in the previous section, cell membrane decorated NP has emerged as a promising method for camouflage by forming a thin layer of the cell membrane coating over the NPs. The camouflaged NPs inherit the properties of the source cells, depending on the source cells used. For example, when RBCs are employed as the source membrane, membrane-coated NPs are found to possess immune evasion and prolonged circulation [42]. Biomimetic NPs attain these cell mimicking properties by the transference of the source cell’s membrane proteins onto the surface of NPs [39]. This functionalization approach is regarded as highly versatile, allowing the delivery of a wide range of cargoes that encompass various inner-core materials.

Targeted drug delivery employs the inherent adhering capability of source cells. For example, NPs camouflaged with a layer of cancer cell membranes showed inherited homotypic adhesion properties, and an intrinsic capacity to bind with the source cells [20,51]. In addition, NPs camouflaged with platelet membranes displayed the ability to mimic platelet binding with pathogens, such as methicillin-resistant *Staphylococcus aureus*, for targeted antibiotic delivery. Meanwhile, platelets help in recognizing tumor cells, including circulating tumor cells, through their ligand binding interactions. Platelet membrane-camouflaged NPs were primarily formulated for the site-specific delivery of anticancer drugs. These persuasive applications inspired the development of cell membrane-camouflaged NPs for targeted antibiotic delivery against the *H. pylori* infection. Angsantikul et al. [60] reported a nanotherapeutic that was obtained by coating antibiotic-loaded poly(lactic-*co*-glycolic acid) (PLGA NPs) with a gastric epithelial cell membrane against an *H. pylori* infection. In their study, it was found that the gastric epithelial cellular membrane-coated NP had the same surface antigens as the source cells that exhibit inherent adhesion towards *H. pylori* bacteria [60].

The use of bacterial membranes as vaccination materials has gained considerable interest. They can stimulate innate immunity and promote adaptive immune responses by exhibiting different pathogen associated-molecular patterns (PAMPs) for a large number of immunogenic antigens with adjuvant properties [83]. Camouflaging NPs with covering bacterial membranes results in the preservation of bacterial characteristics, and thus helps in mimicking natural antigen presentation by bacteria to the immune system. Gao et al. [83] reported a bacterial membrane coated NP for antibacterial therapy, in which gold NPs were coated with bacterial outer vesicles. In this study, they chose *E. coli* bacteria, obtained its outer membranes, and coated gold NPs of 30 nm size with them; they found that this could induce rapid activation and DC maturation in the lymph nodes. Further, vaccination with these NPs produced long-lasting and robust antibody responses [83].

#### 2.1.5. Exosomes

Exosomes are nanosized membrane-enclosed extracellular vesicles originated from the inner endosomal membrane. These vesicles are composed of a lipophilic bilayer with proteins and genetic materials such as micro RNAs, mRNAs, and DNAs [84]. Exosomes are the mediator between cells and can induce immune response by activating natural killer (NK) cells, dendritic cells (DC), and T lymphocytes cells [85]. Various physiological stimuli such as inflammation, oxidative stress and cell growth affects the secretion of exosomes from the cells which is used as a prominent diagnosis marker [86]. Exosomes are acts as vaccination against infection. It can be used as the carrier of pathogen antigens to by modulating the immune response and recruiting monocytes, macrophages, NK cells, and T cells against the infectious agents [87].

### 2.2. Cargoes Used for Immunomodulatory Nanovaccines

As mentioned in the earlier Section 2.1, biomimetic immunomodulatory nanovaccines are composed of 1) biomimetic NPs, and 2) the cargoes used. In this section, the different types of cargoes used for nanovaccines are explained.

#### 2.2.1. Adjuvants

Adjuvants are ingredients used in vaccines to enable the body to produce a stronger immune response, and help vaccines work better. There are different mechanisms by which adjuvants elicit immune responses, which are as follows: 1) prolonged release of antigen at the site of injection, 2) cytokines and chemokine level gets upregulated, 3) recruitment of cells at the injection site, 4) antigen uptake and presentation to antigen-presenting cells increases, 5) APC activates and matures, resulting in the migration to draining lymph nodes, and 6) inflammasome activation [88,89,90]. Generally, adjuvants are classified based on their mechanism of action, physicochemical properties, and origin. Adjuvants can be classified as delivery systems or immune potentiators, depending on their action mechanism. Table 2 describes the partial list of adjuvants used in the abovementioned three categories.

Champion et al. reported a vault NP vaccine for inducing protective immunity at distant mucosal surfaces. These vault NPs contain immunogenic proteins, and hence they are considered as adjuvants [81]. In another study, Riitho et al. [91] formulated a biomimetic vaccine by encapsulating a viral protein inside a polymeric shell, wherein the viral protein was known to have effective cross-presentation by MHC class I. These polymeric NPs were adjuvanted with polyinosinic: polycytidylic acid (poly(I:C)), and loaded with viral proteins that act as antigens. These nanovaccines exhibited significant virus-neutralizing activity, and they were effective against infections caused by the bovine virus diarrhea-virus [91]. Wang et al. [22] reported the use of a dual-functional nanomodulator to enhance CpG mediated cancer therapy. In their work, they synthesized manganese oxide nanosheets and conjugated anticancer drug doxorubicin (DOX) and CpG-silver nanoclusters as the adjuvant [22]. Yang et al. [52] reported the use of a lipid (DSPE-PEG-mannose) modified cancer cell membrane that was coated onto a polymeric NP loaded with adjuvant TLR 7 for an anticancer effect [52]. Recently, Le et al. [92] suggested an in situ nanoadjuvant as a tumor vaccine to prevent the long-term recurrence of tumors. In their study, polydopamine NPs were loaded with imiquimod, and then the NP surface was modified with programmed death-ligand 1 (PDL1) antibodies for the co-delivery of both antigen and adjuvants to the same antigen-presenting cells. This nanoadjuvant with PDL1 antibody could block PDL1 immune checkpoint in tumors, and it is expected to have combinational photothermal and immunotherapy effects [92].

Moon et al. reported the development of a recombinant antigen derived from the circumsporozoite protein, which is the most predominant membrane protein on sporozoites. Their initial work stated that this recombinant antigen, when mixed with conventional antigens could elicit an antigen-specific antibody response [21]. They used a lipid enveloped polymeric NP, and conjugated the malarial antigen into the lipid membrane with an immunostimulatory molecule monophosphoryl lipid A incorporated into the lipid membranes, which resulted in a pathogen-mimicking NP vaccine [21]. Another study suggested that antigen-loaded NPs that display monophosphoryl lipid A (MPLA) and further encapsulation with adjuvant CpG motifs and model antigen Ovalbumin could act as an efficient bacterial vaccine [93]. In that study, CpG potency was found to be enhanced when it was encapsulated inside the NP, which in turn highlights the importance of the biomimetic presentation of pathogen-associated molecular patterns. Because of MPLA with CpG, the pro-inflammatory, antigen-specific T helper 1 (Th 1) cellular and antibody-mediated immune responses were significantly increased [93]. Sahu et al. [23] reported the use of monophosphoryl lipid A (MPLA) NPs loaded with a Hepatitis B surface antigen (HBsAg) for delivery in the colon, which provided prolonged immunization against the Hepatitis B infection. In this study, MPLA was the adjuvant; it activated toll-like receptor type 4 (TLR 4) and Hbs Ag that act as the antigens to be delivered, and thus enabled the simultaneous delivery of both adjuvant and antigens inside the colon. The results indicated that it was effective in the generation of humoral and cellular immune responses [23]. Stimulator of interferon gene (STING) is a prominent agonist which stimulates cyclic dinucleotides (CDNs) to activate IRF3 and NFκB pathways and secrete various pro-inflammatory cytokines. Jack Hu et al. had developed pH sensitive capsid-like hollow polymeric nanoparticle loaded with STING agonist, cyclic diguanylate monophosphate (cdGMP), as a Middle East respiratory syndrome coronavirus (MERS-CoV) vaccine. Delivery of both STING agonist and MERS-CoV receptor binding domain antigen in the surface of the nanoparticle mimicked as virus-liked nanoparticle and induced Th1 type immune response which is a prominent vaccine against the infection [94].

#### 2.2.2. Detained Bacterial Toxins

A toxoid is a chemically or physically modified toxin that is no longer harmful but retains immunogenicity. Wang et al. [20] developed a nanotoxoid that consists of RBC membrane-coated polymeric NPs, and the membrane coating acts as a substrate for the pore-forming staphylococcal α-hemolysin (Hla) nanotoxoid, thereby effectively triggering the formation of germinal centers, and inducing high anti-Hla titers. Further, the nanotoxoid formed showed superior protective immunity against methicillin-resistant *Staphylococcus aureus* (MRSA) skin infection (Figure 3) [20].

Recently, Wei et al. [44] reported a macrophage-membrane-coated nanotoxoid against pathogenic *Pseudomonas aeruginosa*. It has already been reported previously that alveolar macrophages have cationic proteins that can bind to the outer membrane of the bacteria *Pseudomonas aeruginosa,* and its flagella also get involved in phagocytosis.

## 3. Advantages of Nanovaccines

Biomimetic nanovaccines are ideal vaccine candidates, as they have unique physicochemical parameters, such as size, shape, and biomimicking property. This feature makes them a versatile delivery system for the delivery of antigens and adjuvants. The main advantage of nanovaccines is their ability to incorporate both antigens and adjuvants within a single particle to produce maximum stimulation. The biomimicking property of these nanovaccines reduces interactions with RES cells, provides longer circulations, and prevents the burst release of adjuvants from its nano-formulation. The synthesis methods and the choice of material used for NP formulations make the nanovaccine flexible, so that it can incorporate different molecules, such as proteins, polysaccharides, lipids, polymers, and nucleic acids. The NP localization can be enhanced by modifying the NP surface with ligands that have specificity to immune cell receptors [95]. Moreover, antigens and adjuvants can be loaded into NPs either individually or for a combinatorial approach, and protect its molecule integrity from different enzymes, such as nucleases and phosphatases [96]. Besides these advantages, NP formulation also prevents adjuvants from degradation, protects the body from potential systemic toxicity caused by the premature release of adjuvants, and enhances immune response through extended cargo release [19].

Another advantage of biomimetic nanovaccines is their ability to target immune cells; because they are of nanosize, the nanovaccines drain into the lymphatic system, allowing for efficient delivery to lymph nodes, where immune cell density is high [97]. The selection of biomimetic NP plays an essential role in improving vaccine efficiency. Biomimetic nanovaccines helps in shielding the NPs to be recognized from mononuclear phagocytic system and helps in immune escape. Shielding the NPs protects the cargoes from premature release and further modification of the surface of nanovaccines with certain receptors enhances the targeting ability as well as helps in enhanced accumulation [98].

## 4. Applications of Biomimetic Nanovaccines

Biomimetic nanovaccines are developed by the simulation of the synthetic NPs with biologically derived materials, and the combination of both the synthetic and biological properties is the key factor to improve the therapeutic efficacy of the treatment. Biomimetic nanovaccines come in several varieties, such as liposomes, proteins, cell-membrane-coated NPs, and VLPs modified with antigens and adjuvants (shown in Figure 4) for the stimulation of immune responses in our body. Due to the presence of various cell membrane proteins and antibodies on the surface of nanovaccines, it is possible to quickly evade the immune system. Biomimetic surface engineering is an unusual approach towards developing current therapeutic actions.

### 4.1. Anti-Bacterial Therapy

Bacterial infections are marked as life-threatening diseases caused by pathogenic bacteria. To counter these infectious diseases, antibiotics were introduced in the 20th century [99]. The role of antibiotics is to interfere with the growth cycle of the bacteria and suppress the reproduction rate. Antibiotics can disinfect the surface and eliminate bacteria from the body [100]. Overexposure of the antibiotics for a more extended period lowers their effectiveness against infections. NPs can make direct contact with the bacteria cell wall without cell penetration, which shows NPs efficacy as an alternative to antibiotic resistance [22].

Biomimetic NPs have been investigated more as an alternative drug delivery carrier, due to their remarkable blood circulation time, biocompatibility, and targetability. Bacterial membranes stimulate innate and adaptive immunity inside the human body due to the presence of immunogenic adjuvants and antigens, which express numerous pathogens associated with molecular patterns (PAMPs) [61]. Therefore, bacterial-membrane-coated NPs are considered as potential vaccines for antibacterial therapy. Weiwei et al. reported an antibacterial vaccine that showed an effective immune response against pathogens for *Neisseria meningitides* treatments [60]. The functionalization of the Gold NP (size: 40 nm) with the outer vesicle of the bacterial membrane extracted from *E. coli* (BM-AuNPs) showed remarkable serum stability (shown in the Figure 5). Rapid DC maturation in the lymph node and strong antibody response were induced through the BM-AuNPs vaccination. BM-AuNPs produced bacterium specific T-cell response and higher production of interferon-gamma (IFN-γ) and interleukin 17 (IL-17), which is responsible for the Th1- and Th17-based T-cell response against bacterial infection [61].

Wang et al. [4] had reported an anti-virulence biomimetic nanovaccine, assembled with cell membrane coating against methicillin-resistant staphylococcus aureus (MRSA) skin infection. The RBC-membrane coated PLGA NP acts as a natural substrate for pore-forming toxins that can entrap pore-forming staphylococcal α-hemolysin (Hla) onto the surface to reduce MRSA infections [20]. The VLPs vaccine was developed from the Hepatitis B virus core protein with a combination of *Mycobacterium tuberculosis* antigen culture filtrate protein 10 (CFP-10) against tuberculosis (TB). CFP 10 is a T-cell antigen that induces vigorous CTL activity and the secretion of IFN-γ, and it has been reported as a significant TB vaccine. This biomimetic vaccine has expressed antigen-specific Th1 immunity, and is considered as an effective TB vaccine [101]. Endolysins are bacteriophage-secreted enzymes that are responsible for the degradation of peptidoglycan presented in the bacterial cell wall. The liposomal delivery of endolysin is a significant way to treat against gram-positive bacteria. This can overcome the drawbacks of the native endolysin, which is unable to penetrate the outer membrane of the bacteria [102]. RBC membrane-coated biomimetic supramolecular gelatin nanoparticle loaded with vancomycin (Van-SGNPs@RBC) have been developed for the on-demand delivery of antibiotics [103]. The RBC membrane coating provides immune evasion and triggers the accumulation of nanovaccine at the infected site. Due to the RBC membrane coating on the surface, Van-SGNPs@RBC nanovaccine can adsorb bacterial endotoxins and reduce endotoxin-related side-effects in patients. A large number of gelatinases are secreted from the bacteria in an infectious microenvironment. The nanovaccine is responsible for hydrolyzing the gelatin, and triggers the loaded drug (vancomycin) to reduce bacterial infection [103]. The immune-evasion property of Van-SGNPs@RBC was examined by labelling the NPs with Cy5 and incubating them in RAW 264.7 macrophage cells. The results showed that Van-SGNPs@RBC has less macrophage uptake compared to Van-SGNPs, which indicates the circumvention of the Van-SGNPs@RBC NP by immune cells.

### 4.2. Anti-HIV Therapy

Highly active antiretroviral therapy (HAART) is a prominent strategy for the treatment of acquired immunodeficiency syndrome (AIDS) caused by the human immunodeficiency virus (HIV). Nanovaccines, such as nanocapsules, nanocrystals, lipid NPs, nanocarriers, liposomes, and micelles, have been recently investigated for anti-HIV therapies. Although many anti-retroviral drugs are available for treatment, none of them can eradicate the viral reservoir [104]. Engineered biomimetic nanovaccines have adverse properties in modulating our immune system against viral infection. They show high encapsulation efficiency of anti-retroviral drugs, cytokines, and enzymes and site-specific drug releases [105]. Liposomes are endocytosed through mononuclear phagocytic system cells (MPS), and reach the HIV-infected reservoir [106]. The liposomal delivery of anti-HIV vaccines was investigated to induce antibody and cellular immune responses 25 years ago [107]. The formulation of the liposomal vaccine with IL-7 immune stimulator and recombinant HIV envelope protein (env-2-3SF2) as an antigen showed a strong antibody response, compared to liposomal delivery with IL-7 or the liposome alone. When pathogen-free mice were vaccinated with env-2-3SF2 and IL-7, the antibody production and CTL activity were significantly increased [107]. Hanson et al. [108] investigated the liposomal delivery of membrane-proximal external region (MPER) with a suitable antigen, monophosphoryl lipid-A (MPLA), and the stimulator of interferon gene (STING) agonist cyclic-di-GMP (cdGMP) as an active HIV vaccine. The administration of the liposomal vaccine with MPER, molecular adjuvants MPLA and cdGMP achieved a significant humoral response, as well as T-cell responses [108]. Andersson et al. [109] developed HIV VLPs composed of HIV env antigen HIV_BaL_ gp120/gp41, which is a viral surface glycoprotein that targets HIV-1 and TLR ligands. These HIV VLPs vaccines modulate the immune system and maintain the germinal center from B-cell hypermutation. Preservation of the germinal center results in the secretion of high HIV neutralizing antibodies. VLPs, like nanovaccines composed of antigen and different TLR ligands, such as TLR2 (PAM3CAG), TLR3 (dsRNA), TLR4 (MPLA), or TLR7/8 (resiquimod), can accelerate the immunogenicity of mice. This combined nanovaccine induced Th-1 like cytokines, and prolonged the lymph node germinal center and T follicular cells for antibody production [109]. Intranasal immunization of the HIV VLP nanovaccine for 12 weeks reduced the env-specific IgG1 titers. However, IgG2b, IgG2c, and IgG3 titers were well maintained during the study, which is the main factor for generating a neutralizing antibody against HIV. The combination of both antigens and adjuvants with VLPs showed a robust immune response and well-maintained germinal center. VLPs encoded with the HIV-1 adenovirus primed immunogen is an effective strategy towards HIV vaccination. The envelope glycoprotein (Env) is an antibody-inducing prophylactic drug presented on the HIV-1 particle. VLPs encoded with docked HIV-1 consensus Env antigen produced the antibody response, and released more neutralizing antibodies against HIV [110]. The ectodomain protein, gp140, has been investigated recently as an alternate Env targeting for induction of neutralizing antibodies against HIV [111]. Delivery of lipid nanocapsule modified with trimeric gp140 (gp140T) on the surface had promoted a strong antibody response and timer-antibody binding. Composition of nanocapsule and gp140T induced a remarkable humoral response over 90 days against Env immunogen compare to soluble trimer adjuvanted protein in oil-in-water emulsion [112].

Cell-membrane-coated NPs have also emerged as an effective platform to treat HIV infections. HIV infection explicitly targets T-cells, and reduces the immune cells by viral killing, where uninfected cells lead to the apoptosis. Wei et al. [47] developed a T-cell membrane-coated biomimetic nanovaccine to neutralize the viral infection (shown in Figure 6). The viral fusion of the virus and immune cells is started by the interaction between the CD 4 receptor and the glycoprotein (gp120) through the C-C chemokine receptor 5 (CCR5) and C-X-C chemokine receptor type 4 (CXCR4) [113]. The T-cell modified PLGA nanovaccine was mimicked as a parent T-cell for inducing the specific binding to HIV. This biomimetic agent diverted the viral attack, and depleted the viral infection [47].

### 4.3. Anti-Malarial Therapy

Malaria is a ubiquitous parasite disease found worldwide and is caused by protozoan parasites. The current treatment for malaria involves the oral administration of the traditional antimalaria drugs, such as chloroquine, pyrimethamine, artesunate, and sulfadoxine. But the potency of these drugs is diminished by the drug resistance ability of these parasites. The downside of the current treatments for malaria includes low stability in the stomach, higher side effects, and low half-life inside the body [114]. Nanovaccines are the best alternatives to combat this parasite disease. Nanocarriers can carry active drugs to specific sites with minimal loss and side effects for adverse therapeutic effects [114]. Biomimetic nanocarriers, such as liposomes and proteins, are highly biocompatible and promising for the drug delivery application [115]. Malaria vaccines are less resistant against recombinant antigens and require repeated re-boosting. Liposomes are a well-known drug carrier that can deliver the drug within the host without degradation [116]. The surface modification of a liposome with the targeting ligands and antibodies can precisely bind to the infected cells and facilitates site-specific drug delivery. Marques et al. [117] reported that heparin-coated liposomes, loaded with primaquine, had an adverse antimalarial activity. Due to the higher binding affinity of the heparin towards the heparin-binding protein in the infected erythrocyte cell membrane surface, it delivered the drug to infected sites. Immunoliposome (ILP), a liposome modified to target the immune system, has been recently investigated for antimalarial activity to target plasmodium-infected red blood cells (pRBC) [118]. Liposomes modified with glycosaminoglycan chondroitin 4-sulfate (a heparin substitute) for the delivery of primaquine have shown an additive effect compared to the control [119]. *Plasmodium falciparum* erythrocyte membrane protein 1, the primary receptor for chondroitin-4 sulfate, is a parasite-mediated antigen that is present in the endothelium of postcapillary venule. It enhances the adhesion of the liposomes towards pRBC [119]. Rajeev et al. [120] reported an antimalaria vaccine by liposomal delivery of merozoite surface protein (MSP-1) which is presented on the surface of *Plasmodium falciparum*. The transcutaneous injection of this antigen accelerated immune responses by activating epidermal antigen-presenting cells. The liposomal delivery of membrane antigen induces strong humoral and cell-mediated immune responses [120]. Labdhi et al. [121] developed a self-assembled protein nanovaccine delivery with adjuvant-based liposomes to target *Plasmodium falciparum.* The self-assembled protein NP contained 60-identical monomer protein chains comprised of *P. falciparum* Circumsporozoite Protein (P*f*CSP), CD 4+, CD 8+, and T_H_ epitopes for inducing immune responses. Adjuvant-augmented (QS21, alhydrogel) liposomal delivery of the self-assembled protein nanovaccine targeted the native P*f*CSP and stimulated the immune responses, with 80% or more mice gaining complete protection from malaria [121]. The delivery of two antimalarial drugs, such as lipophilic aminoquinolines and amino alcohol derivative encapsulated into ILP, had more than 90% encapsulation efficiency through a citrate buffered pH gradient method. The ILPs performed in vivo RBC targeting, showed higher retention time, and reduced malaria parasite densities in blood, compared to the non-targeted delivery [121]. RTS,S vaccine, developed by GlaxoSmithKline (GSK), was the first malaria vaccine for clinical trials [122]. RTS, a single polypeptide that is specific towards Plasmodium falciparum, is fused with S polypeptide to produce VLPs. RTS, S antigen, and AS01 was an effective formulation to reduce 46% malaria infection in children [122]. RTS,S vaccine is circumsporozoite protein (CSP)-based VLPs from the CSP-hepatitis B surface antigen (HBsAg) fusion protein that targets the pre-erythrocytic stage of *Plasmodium falciparum* infection. Kathrine et al. [123] developed a more immunogenic CSP-based particle vaccine compare to RTS,S, which is named R21. R21 is comprised of a single CSP-hepatitis B surface antigen (HBsAg) fusion protein, which has a higher proportion of CSP than RTS,S, to induce a robust immune response against *Plasmodium falciparum* infection. A low dosage of R21 delivery with adjuvants such as Abisco-100 and Matrix-M achieved strong humoral and cellular immune responses against the sporozoite challenge in BALB/C mice [123]. The combination of thrombospondin related adhesion protein (TRAP) and R21 induced high levels of TRAP-specific CD8^+^ T-cells and is currently under clinical trials.

### 4.4. Anti-Tumor Immunity

The development of biomimetic NPs with chemical and structural modifications to mimic the biological environment is an established approach for cancer therapy. Nanovaccines are novel platforms for delivery of both adjuvants and antigens that generate a strong antitumor response by modulating the immune system [124]. Various type of nanovaccines, such as liposomes, protein NPs, and cell-membrane-coated nanomicelles, have been recently developed for successful anti-cancer therapy [125,126]. The modification and surface functionalization of the biomimetic nanovaccines can accelerate therapeutic activity with high cellular uptake, prolonged circulation, site-specific accumulation, and stimuli-responsive drug releases. Phospholipids are the primary elements for liposome formulation, which mimics a biological membrane [127]. The formulation of an ILP by introducing specific antibodies and antigens onto the surface can induce active targeting and immune modulation [128]. Antigens presented in the liposomes induced immunogenicity inside the body. Encapsulated or surface modified antigens altered the T-cell responses and stimulated the CD4^+^ and CD8^+^ T-cells to fight the tumor. Phosphatidylserine conjugated liposomes are effective vaccines that are significantly captured by antigen-presenting cells, and are responsible for T_h_-cell proliferation [129]. Polyinosinic: polycytidylic acid (Poly I:C) mediated cationic liposome was reported as an adequate vaccine delivery against a natural epitope of HER/Neu-derived P5 peptide that enhances anti-tumor immunity. Poly (I:C) is a TLR 3 agonist that displays strong immune response and triggers apoptosis. The liposomal vaccination of both P5 peptide and Poly(I:C) significantly induced an antitumor immune response by releasing a higher number of CD8^+^ T-cells and interferon-gamma, compared to a single vaccination of either P5 peptide or Poly (I:C). Liposomal injection with P5 and Poly (I:C) induced a strong cytotoxic T lymphocyte (CTL) response, and inhibited tumor growth, compared to other controls [130]. P5 peptide conjugated liposomal delivery of monophosphoryl lipid A (MPLA), an TLR 4 agonist, enhances the secretion of IFN-γ and CTL response by inducing CD8^+^ T-cells. Liposomal vaccination with P5 and MPL achieves significant tumor inhibition and longer survival time [131].

Cell-membrane-coated NPs are other types of biomimetic nanovaccines that are used as anticancer therapeutic agents [126]. Because of the presence of several functional molecules on the cell surface, the cell membrane coating on the nanovaccines acts as a native antigen for the immune cells in the tumor [126]. A cell membrane coating on the hydrophobic core of NPs demonstrated a self-recognizing property for targeting [39]. RBC-coated NPs could evade the immune system, due to the presence of various immunomodulatory markers on the membrane surface, and prolong the circulation for a longer time, thus enhancing the therapeutic activity [103]. Mushi et al. developed a camouflaged nanocarrier coated with Hela human cervix carcinoma cellular membrane onto the nanocarrier. The nanocarrier was composed of doxorubicin, and PD-L1 siRNA loaded into the PLGA NP to target cancer cells [132]. The hybridization of both the cancer cell and RBC membranes is a superior approach to the delivery of therapeutic active molecules for cancer therapy [133]. The cancer cell membrane helps in homo-typing targeting by self-recognition, while the RBC membrane prolongs the blood circulation by evading the immune system of our body [133]. The surface proteins of the cancer cell membrane act as a tumor antigen that will trigger the immune response. Yang et al. [52] reported a cancer cell membrane-coated PLGA NP loaded with TLR-7 agonist, imiquimod (R837), and modified with mannose by a surface lipid anchoring method (Figure 7). Mannose modification on the surface of the nanovaccine can trigger the antigen-presenting cells uptake and lymph node migration for higher DC maturation. Cancer cell membrane coating performed as a targeting moiety and a cancer-specific antigen. Immune modulatory agent imiquimod (R837) can stimulate the production of cytotoxicity T-cells to kill the cancer cells, and the combined biomimetic vaccines act as an anticancer vaccine by inhibiting cancer cell progression, compared to other controls [52]. Melanoma cancer cell-membrane-coated PLGA NP loaded with CpG oligodeoxynucleotide augmented the anti-tumor immunity and could be used as an antigen/adjuvant vaccination. The cancer cell membrane acted as a tumor antigen and enhanced the immune response. This biomimetic nanovaccine triggered the antigen-presenting cell maturation and proinflammatory cytokines, i.e., interleukin-6 and interleukin-12 (IL-12), by modulating the immune responses to cancer cells [134].

Various types of targeting peptides, nucleic acids, and proteins are introduced during the formulation of biomimetic nanovaccines to activate the immune system. CpG oligonucleotides act as a TLR adjuvant, because its recognition by endosomal TLR9 boosts the immune activities against regulatory T-cells inside cancer patients [22]. Antibodies are found to be more effective at targeting specific antigens and over-expressed receptors on cancer cells for enhanced anticancer therapy [135]. Antibodies, such as anti-PD-1, can inhibit the PD-1/PD-L1 pathway, and block the conversion of the cytotoxic T-cell to the regulatory T-cells. A gas-generating liposome loaded with sodium bicarbonate (NaHCO_3_) causes more cell death, and releases a high amount of tumor-associated antigens (TAA) [136]. Combined treatment with a gas-generating liposome and anti PD-1 remarkably enhanced the recruitment of immune cells and CTL responses along with the reduction in regulatory T-cells, compared to single treatment of either anti PD-1 or liposome [136]. The VLPs were developed by using noninfectious viral proteins and capsids to target therapeutic agents. VLPs acted as pathogen-associated molecular parents (PAMPs) to induce immune stimulation against cancer. Lizotte et al. reported that the self-assembled VLPs from cowpea mosaic virus served as effective vaccination, and delayed the tumor growth in B16F10 melanoma mice model [137]. VLPs obtained a durable and long-lasting humoral immune response, and were easily adaptable towards pathogenic threats [138]. Patel et al. reported influenza VLPs modified with breast cancer human epidermal growth factor receptor 2 (HER-2) antigen as a potential therapeutic vaccination against HER-2 expressing tumor. VLPs immunization with the HER-2 antigen enhanced the Th1 and Th-2 type antibody responses, and inhibited tumor growth [139]. VLPs derived from the cowpea mosaic virus are a potent vaccine against mouse ovarian cancer [140]. It induced intra-tumoral cytokine responses by upregulating IL-6 and IFN-γ and downregulating IL-10, which then repolarized the tumor-associated macrophages and neutrophils. The in situ vaccination of this VLP significantly enhanced the tumor-specific CD8^+^ T-cell responses against the aggressive ovarian tumors [140].

### 4.5. Anti-Melittin Therapy

Melittin is a linear cytosolic peptide that is secreted from honey bee venom. If injected into an animal body, it causes pain sensation, owing to pore formation in epithelial cells. This cationic peptide is responsible for cell membrane lysis caused by high interaction with negatively charged phospholipids, and it inhibits ion transportation into cells. The current strategies to fight against melittin are based on toxoid vaccination. The elimination of toxicity in pore-forming toxins and the preservation of the immunity epitope is a considerable challenge for researchers. Biomimetic nanovaccines are the best alternatives for the delivery of these toxins as a suitable toxoid vaccine, since they maintain the antigenic activities of the native toxin to induce an immune response in the body. The RBC membrane-coated NP was used as a suitable carrier to anchor the staphylococcal a-hemolysin (Hla) model toxin towards non-disruptive nanotoxoid formation [25]. Nanotoxoid vaccination stimulated the host body immunity and eliminated the toxins through antigen-presenting mechanisms. Kang et al. [141] reported that the nanotoxoid formation method might be a promising approach for pore-forming toxins (PFTs) vaccines. The synthetic PDA NP can efficiently neutralize melittin to reduce the toxicity of melittin. The interaction of polydiacetylene (PDA) NP with melittin was mediated by both hydrophobic and electrostatic interactions. Melittin-loaded PDA NP was used as a nanotoxoid vaccination to enhance immune activity against melittin. PDA-melittin demonstrated 70% cell viability in DCs, whereas free melittin showed 90% cell apoptosis [141]. This biomimetic nanovaccine maintained the antigenic determinant of the melittin, which was responsible for high DC maturation and cellular uptake. After three doses of biomimetic nanotoxoid vaccination, the mice received the lethal bolus toxin. Biomimetic nanotoxoid vaccinated mice showed a 75% survival rate, compared to the 20% survival rate of non-vaccinated mice [141]. A biomimetic nanosponge was reported (Figure 8) by using PLGA NPs as a core, and RBC membrane as a surface coating. The RBC membrane acts as a substrate for PFTs, which can induce an alpha-toxin onto the surface, reduce hemolytic activity, and enhance the blood circulation time [142].

### 4.6. Foot-and-Mouth Disease Virus Therapy

Foot-and-mouth disease (FMD) is a highly infectious disease caused by FMD viruses found in cloven-hoofed animals; it is transmissible from animal to human [10,143]. FMDV vaccination is the traditional approach, which is time-consuming and expensive. It is less useful to induce sufficient mucosal immunity against the FMDV. NPs are a well-known drug carrier for antigens and adjuvants and can produce strong resistance. Teng et al. [144] developed an FMD vaccine by using gold nanostars (AuSNs) and FMD VLP (FMD VLPs-AuSNs) complexes. FMD VLPs-AuSNs nanovaccines (shown in Figure 9) augmented a robust immune response against FMDV. This biomimetic vaccine manifested a high cellular uptake, due to VLP modification. The macrophage activation with FMD VLPs-AuSNs was relatively higher than VLPs, because AuNPs and the nitric oxide production induced a robust immune activation against the FMDV. Another biomimetic nanovaccine was formulated by using synthetic peptide derived from the FMDV and gold NP (AuNP) [143]. The synthetic peptide of the capsid protein (VP1) of the FMDV had a strong immune activation in guinea pigs with 40% higher efficacy, compared to that for the FMD vaccine.

Table 3 summarizes the type of biomimetic NPs and the therapeutic cargoes used along with its application in the treatment of various diseases.

## 5. Challenges and Future Directions of Biomimetic Nanovaccines

Although nanovaccines are the prominent model to treat various diseases, their efficacy to modulate our immune response against diseases can be excepted more. Some of the disadvantages of biomimetic nanovaccines are based on their stability purposes [150]. Liposomes storage becomes a major drawback as it leads to aggregation and structural destabilization [151]. Scaling up of nanovaccines is also a significant challenge due to its stability as well as cost-effective production in an efficient manner from batch to batch. Multiple loading of different components like antigens and adjuvants in a single nanoplatforms is difficult and becomes more challenging. However, these demerits of the biomimetic nanovaccines can be subjugated with appropriate ideas and advanced technology.

The application of biomimetic nanovaccines is a remarkable evolution in the field of medicine. Biomimetic nanovaccines are the prime attraction of researchers due to its notable advantages and impressive research outcomes that have already achieved. Well established knowledge regarding physiological and immunological behavior of the diseases is the base to design a strong vaccine where biomimetic nanovaccines stand out first. It has the potency to deliver specific on-site delivery, prolonged circulation, reduced side-effects and induction of robust immune response. Finally, nanovaccines based on biomimetic principle have noticeable advantages like biocompatibility, low toxicity, bioavailability, and targetability leads as a prominent agent to treat various diseases.

## 6. Clinical Aspects of Biomimetic Nanovaccines

Very few vaccine candidates have successfully reached the clinic after preclinical evaluations. Most vaccines that are available now in the market can elicit only humoral responses, thereby availing the need for the development of vaccines that can generate strong cellular responses for certain infectious diseases and cancer. One of such biomimetic nanovaccines is “Mosquirix”, which was proved to be effective against malaria. This nanovaccine constituted the circumsporozoite protein of *Plasmodium falciparum* and MPLA 4 with a saponin adjuvant QS-21 [152]. Another nanovaccine, which is currently under clinical trials as “Vaxfectin^®^”, is cationic liposomal formulation by encapsulating therapeutic DNA vaccines against the herpes simplex virus type-2 (HSV-2). Vaxfectin^®^ nanovaccines are also used for DNA immunization against influenza virus H5N1, and are also under clinical trials [153]. Another FDA-approved nanovaccine is Inflexal^®^V, where the HA surface molecules of the influenza virus are directly fused with lipid components, and used as a subunit influenza vaccine [154]. Generalized modulus for membrane antigens (GMMA) was derived from the outer membrane of genetically modified gram-negative bacteria. It can produce Penta-acylated lipopolysaccharide, and these vaccines were used against bacterial infection Shigellosis, and are in clinical trials now [155]. In addition to the nanovaccines, as mentioned earlier, Stimuvax^®^ is another therapeutic liposome vaccine against cancer. It has a lipo-peptide called Tecemotide, which is used as an antigen target specific tumor antigens. However, this vaccine failed in the III phase of clinical trials [156]. Another liposomal therapeutic vaccine, which is a modified form of Stimuvax^®^, is currently under clinical trials; this nanovaccine is composed of a synthetic peptide (antigen), an MPLA immunoadjuvant, and lipids [157]. Another biomimetic nanovaccine is Epaxal, a viral liposomal nanovaccine that uses viral glycoprotein fused with lipids as an adjuvant, and that is used against hepatitis A infection [158].

## 7. Conclusions

Nanovaccines have attracted tremendous interest over the past few years, due to their unique physicochemical characteristics. The roles of nanovaccines as potent vaccine have been examined to boost their therapeutic activity by enhancing their stability, prolonging their circulation and site-specific accumulation, increasing their delivery according to various biological and external stimulus, and overcoming all physiological barriers. As an active immunogenic material to modulate the immune response, nanovaccine enables antigen stability, enhances antigen processing and immunogenicity with targeted delivery, and prevents the burst release of antigens and adjuvants. Nanoscale delivery vehicles help in the design of nanovaccines that can elicit potent immune responses to overcome tumor immunosuppression. Biomimetic nanovaccines have emerged as a promising candidate with multiple functionalities in a single nanoplatform. Biomimetic nanovaccines enable the co-delivery of both antigen and adjuvant in a single platform with minimal side effects, and the camouflaging property of bio membranes makes it noteworthy. Biomimetic nanovaccines can act as a vaccination against various infectious diseases. Due to their intrinsic properties, bio-inspired nanovaccines act as immune-modulatory agents to stimulate DC maturation and cytotoxic T-cell production. As a bio-carrier, nanovaccines can transport both antigens and adjuvants with active drugs to enhance antigen presentation and immune response activation. Biomimetic nanovaccines are suitable epitopes to produce adequate antibodies, such as neutralizing antibodies, against viral and parasite infection. They act as natural substrates to adsorb the endotoxins onto their surface to relieve the infections in our body. Tumor suppressive agents are re-challenged by the nanovaccines, and the response against the cancer cell is being attuned to eradicate it. Notwithstanding a few challenges and limitations to biomimetic nanovaccines, the advantages as mentioned above demonstrate that these nanovaccines will conquer and open various novel therapeutic modalities for various diseases.

## Figures and Tables

**Figure 1 pharmaceutics-11-00534-f001:**
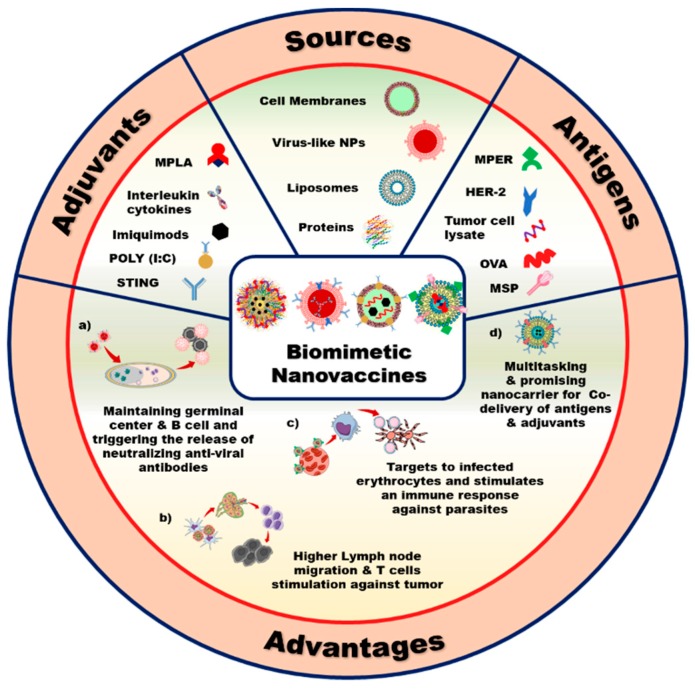
Schematic of different formulations of biomimetic nanovaccines and their advantages. **a**) Biomimetic nanovaccines maintain the germinal center and B-cells inside our body, which are responsible for the release of antiviral neutralizing antibody against viruses, **b**) biomimetic nanovaccines strengthen the humoral immune response by inducing higher DC maturation and stimulating cytotoxic T-cell to kill cancer cells, **c**) biomimetic nanovaccines can target infected blood cells, and induce a strong immune response inside our body, and **d**) biomimetic nanovaccines are suitable candidates for carrying antigens, adjuvants, and therapeutic molecules. (MPLA: monophosphoryl lipid A, STING: stimulator of interferon gene, POLY (I:C): polyinosinic:polycytidylic acid, MPER: membrane-proximal external region, HER-2: human epidermal growth factor receptor 2, OVA: ovalbumin and MSP: merozoite surface protein).

**Figure 2 pharmaceutics-11-00534-f002:**
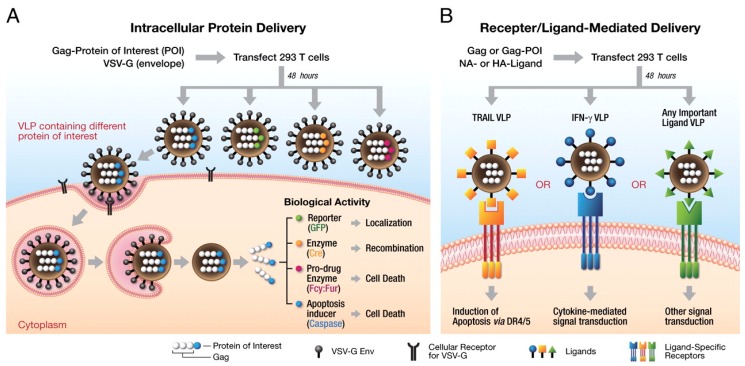
Schematics of the generation of functional virus-like particle (VLP) and (**A**) the delivery of proteins of interest intracellularly, and (**B**) by receptor/ligand-mediated protein delivery. Reproduced with permission from Ref. [57]; Copyright © 2011, National Academy of Sciences.

**Figure 3 pharmaceutics-11-00534-f003:**
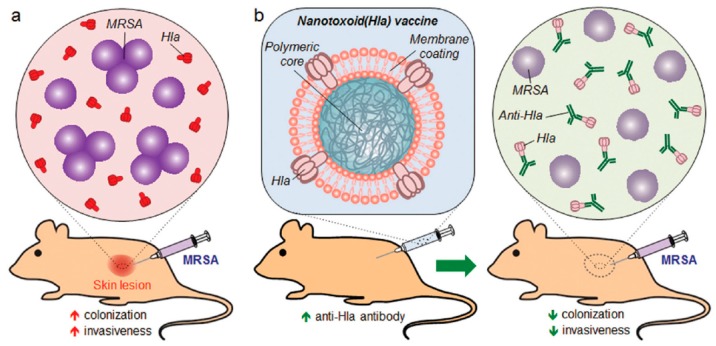
Scheme of biomimetic nanotoxoid showing protection against methicillin-resistant *Staphylococcus aureus* (MRSA)-induced skin infection. **a**) The normal condition of skin lesion formation in which MRSA bacteria employs hemolysin (Hla) and helps in colonizing the site. **b**) After nanotoxoid vaccination, anti-Hla and neutralize the toxins produced by MRSA. Reproduced with permissions from Ref. [20], Copyright © 2016, John Wiley and sons.

**Figure 4 pharmaceutics-11-00534-f004:**
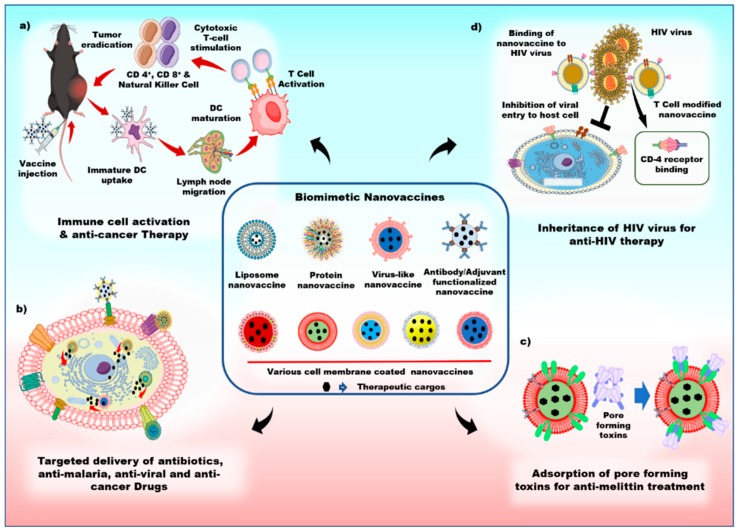
Schematics of the various applications of biomimetic nanovaccines. **a**) Biomimetic nanovaccines with adjuvants, antigens, and antibodies can trigger dendritic cell maturation and stimulate cytotoxic T-cell to induce a strong immune response against tumor, **b**) biomimetic nanovaccines can actively target the cancer cell, and effectively deliver the therapeutic drugs, **c**) biomimetic nanovaccines act as a natural substrate for the adsorption of pore-forming toxins, and **d**) biomimetic nanovaccines can bind to human immunodeficiency virus (HIV) viruses with CD-4 receptors, and prevent the host cell from HIV virus infections.

**Figure 5 pharmaceutics-11-00534-f005:**
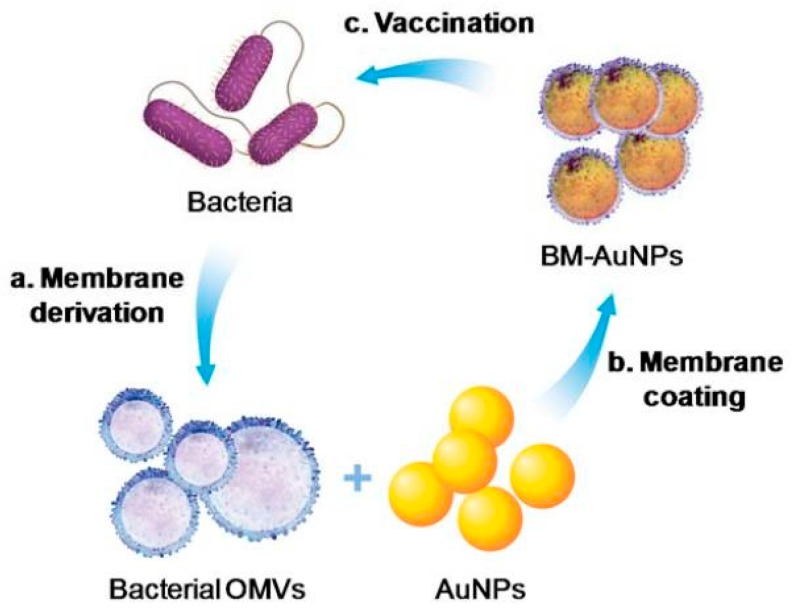
Schematic of antibacterial modulation via bacterial membrane-coated nanoparticles (NPs). Reproduced with permission from Ref. [61], Copyright © 2015, American Chemical Society.

**Figure 6 pharmaceutics-11-00534-f006:**
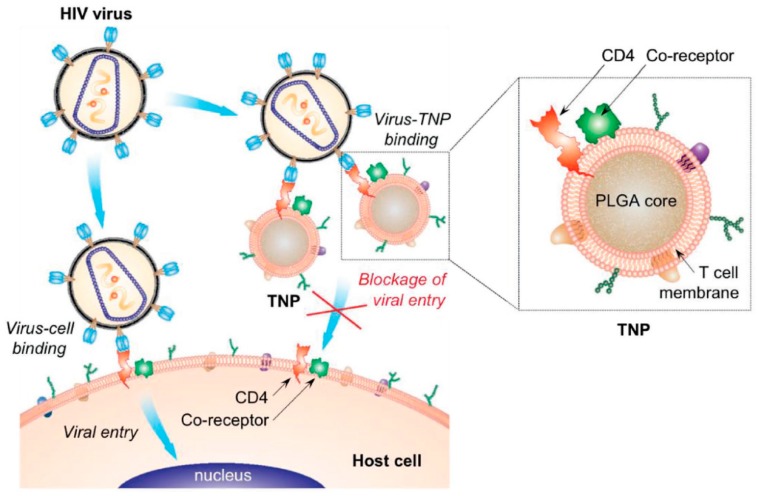
Schematic of T-cell membrane coated nanoparticle (NP)-mediated depletion of HIV infection. Reproduced with permission from from Ref. [47], Copyright © WILEY-VCH Verlag GmbH & Co. KGaA, Weinheim.

**Figure 7 pharmaceutics-11-00534-f007:**
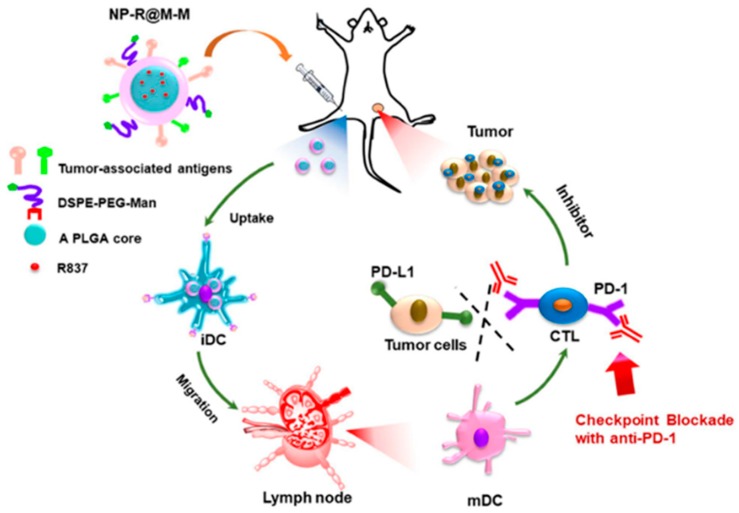
Schematic of the cancer cell membrane-coated, R837 loaded, and mannose modified poly(lactic-*co*-glycolic acid) (PLGA) nanovaccine for anticancer vaccination. Reproduced with permission from Ref [52], Copyright © 2018, American Chemical Society.

**Figure 8 pharmaceutics-11-00534-f008:**
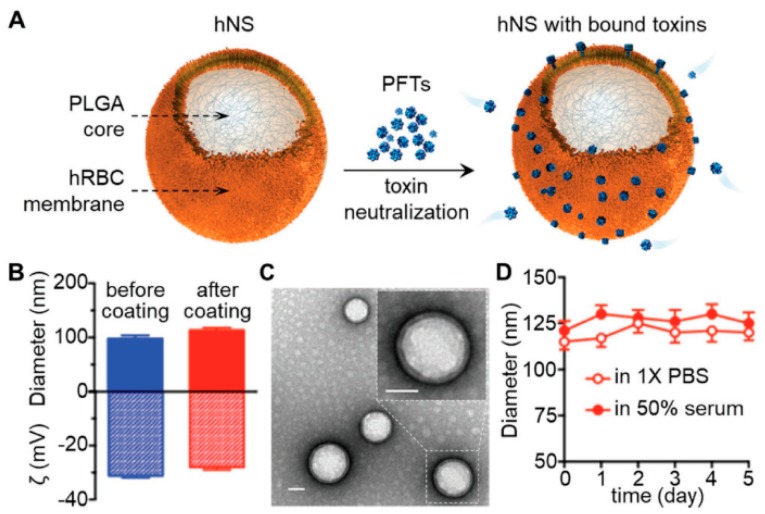
(**A**) Schematic of the biomimetic nanosponges and neutralization of the pore-forming toxins (PFTs) mechanism. (**B**) Dynamic light scattering measurement of the NP hydrodynamic size and zeta potential. (**C**) TEM image of the nanosponge. (**D**) Stability of the NP. Reproduced with permission from Ref. [142], Copyright © 2018 WILEY-VCH Verlag GmbH & Co. KGaA, Weinheim, published Feb 13, 2018.

**Figure 9 pharmaceutics-11-00534-f009:**
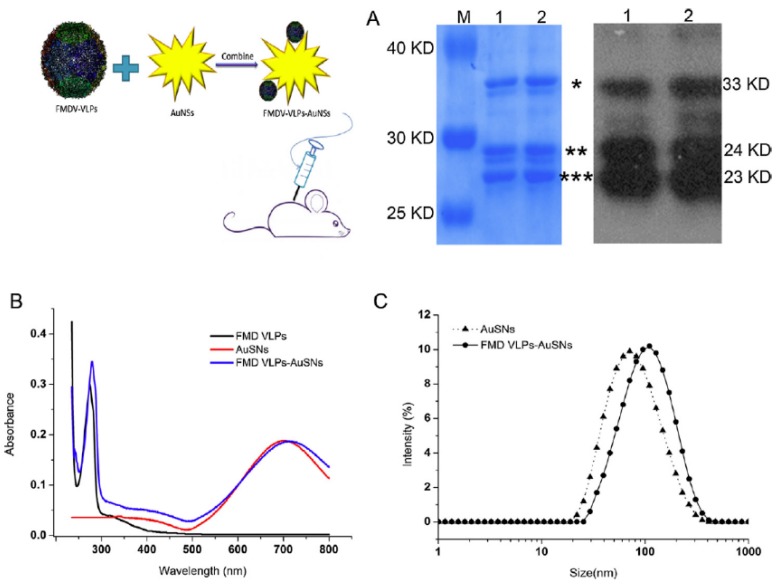
**A**) Schematic of the preparation of foot-and-mouth disease virus-like particle gold nanostars (FMD VLPs-AuSNs) complex. SDS page and western blot analysis FMD VLPs and FMD VLPs-AuSNs complexes; **B**) Fluorescent absorbance of the nanocomplex. **C**) Size distribution. Reproduced with permission from Ref. [144], Copyright © 2018 Elsevier Ltd. All rights reserved.

**Table 1 pharmaceutics-11-00534-t001:** Already reported biomimetic nanovaccines and its applications.

Nanoparticles	Components	Application	References
Liposomes	Liposome-polycation-DNA NPs	DNA vaccine delivery	[24]
	PLGA NPs with lipid antigens	Malarial vaccine delivery	[21]
	Cancer cell membranes with lipids coated onto polymeric NPs	TLR 7 delivery: Anticancer vaccine	[52]
VLPs	Avian retrovirus with Gag fusion proteins	Intracellular protein delivery	[57]
	Genetically modified VLP	Anti-viral protection	[58]
Self-assembling proteins	Hollow vault protein	Suppress lung cancer proliferation	[59]
Cell membrane decorated NPs	Gastric epithelial cell membrane coated PLGA NPs loaded with antibiotics	Anti-bacerial therapy	[60]
	Bacterial membrane coated Gold NPs	Antibacterial immunity	[61]

**Table 2 pharmaceutics-11-00534-t002:** Few types of adjuvants used and their classification.

Immune Potentiators	Delivery Systems
dsRNA: Poly (I:C), Poly-IC:LC MPLA (monophosphoryl lipid A) LPS (Lipopolysaccharide) CpG oligodeoxynucleotides Flagellin Imiquimod (R837) Resiquimod (848) Saponins (QS-21)	Aluminum salts Incomplete Freund’s reagents Virus-like particles Polylactic acid, Poly(lactic-*co*-glycolide) data

**Table 3 pharmaceutics-11-00534-t003:** List of biomimetic nanovaccines for various treatments.

Type of Biomimetic Nanoparticle (NP)	Therapeutic Cargo	Application	Reference
Liposome	Hepa 1-6 cell lysate and Poly I:C	High tumor specific CTL immune response	[145]
	P5 peptide and Poly I:C	CTL immune response and anti-cancer therapy	[130]
	Tumor associated ESO-1 antigen and IL-1, MAP-IFN-γ	Fcγ receptor targeting and anti-cancer therapy	[146]
	OVA antigen	CTL response and cancer immune therapy	[147]
	Endolysin	Degradation of bacterial protein and anti-bacterial therapy	[102]
	Env-2-3-SF2, IL-7	Strong antibody response and anti-HIV therapy	[107]
	MPER and MPLA, STING, cdGMP	Strong T-cell response and anti-HIV therapy	[108]
	MSP-1	Activation of epidermal APC	[121]
Virus like NP	CFP 10	CTL activity, Th1 immune response, and anti-bacterial therapy	[101]
	HIV env antigen	Maintaining the germinal center, and releasing neutralizing antibody for anti-HIV therapy	[110]
	CSP-hepatitis B surface antigen and Abisco-100, Matrix-M	Targeting infected erythrocytes and CD8 + T-cell responses in anti-malaria therapy	[123]
	HER-2 antigen	Th1 & Th2 type antibody response and anti-cancer therapy	[139]
Outer membrane coated nanovaccine	Alum adjuvant	Th2 type immune response anti-bacterial therapy	[148]
RBC membrane coated Nanovaccine	None	Adsorption of bacterial endotoxin	[149]
	None	Natural substrate for PFT for Anti-melittin therapy	[142]
T-cell coated Nanovaccine	None	Inhibition of viral attack to host cell	[113]
Cancer cell membrane coated Nanovaccine	PD-L 1 siRNA	Tumor targeting and anti-cancer therapy	[132]
	CpG oligodeoxynucleotide	Stimulation of APC maturation and release of pro-inflammatory cytokines in anti-cancer therapy	[134]

(Poly I:C -Polyinosinic:polycytidylic acid, ESO 1- esophageal cancer, MAP-IFN-γ - multiple antigenic peptide- interferon -γ, Env- envelope glycoprotein, MPER- membrane-proximal external region, MPLA- monophosphoryl lipid A, STING- stimulator of interferon gene, cdGMP- cyclic-di-GMPHER-2: human epidermal growth factor receptor 2, OVA: ovalbumin and MSP: merozoite surface protein, CFP-10- culture filtrate protein 10).

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
