# Peer review of "Recent Advances in Nanovaccines Using Biomimetic Immunomodulatory Materials"

_pharmaceutics, 2019, doi:10.3390/pharmaceutics11100534_

Round 1

Reviewer 1 Report

In the review article “Recent Advances in Nanovaccines Using Biomimetic Immunomodulatory Materials”, the authors present recent advances in biomimetic nanoparticle-based vaccine including liposomes, polymeric nanoparticles (NPs), cell-membrane coated NPs, self-assembling protein NPs, and virus-like particles. They discuss the use of nanovaccines in anti-bacterial therapy, anti-HIV therapy, anti-malarial therapy, anti-melittin therapy, and anti-tumor immunity. Moreover, the authors explain about the cargoes used for nanovaccines (different types of adjuvants, and bacterial toxins). Overall, this manuscript is well written and informative. There are several points that could be addressed to further improve the manuscript as below.

In the “Liposomes” section, the authors mention the concept about “virosome”. Reviews on influenza-derived virosomes(Expert Review of Vaccines, 10(4), 437-446, 2011) could be added in this section for audience who might be interested in the topic.

In “2.1. Adjuvants”, the stimulator of interferon genes (STING) and its agonists are an emerging class of adjuvant targets worth discussing as it is a primary innate immune signaling pathway triggered by many bacterial and viral pathogens. Among biomimetic nanoparticle vaccines, a viromimetic STING-agonist loaded nanoparticle vaccine for MERS-CoV vaccination is worth discussing in the review. (Advanced Functional Materials, 29(28), 1807616, 2019)

In “2. Anti-HIV therapy”, studies on nanoparticle vaccines anchored with His-tagged gp140 trimers as a potential HIV vaccine formulation for inducing anti-HIV humoral responses are worth including (Bioconjug Chem, 25(8), 1470-1478, 2014).

Author Response

We express our sincere gratitude to the reviewers for their comprehensive and valuable comments. We revised the manuscript according to reviewers’ guidance and the explanations are given below. We have attached the main text file indicating the changes highlighted in red color.

Reviewer 2 Report

I read with interest the manuscript entitled "Recent Advances in Nanovaccines Using Biomimetic Immunomodulatory Materials" by Vijayan et al. In this review the authors described the biomimetic nanovaccines in terms of composition, use and efficacy in different therapeutic settings, such as anti-bacterial therapy, anti-HIV therapy, anti-malarial therapy, anti-melittin therapy and anti-tumor immunity.

The paper is accurately written and exhaustive, although too long and redundant in some aspects. I would suggest to include more tables to summarize the information described in the text.

There are some points that should be addressed:

-the description of “types of biomimetic NPs” (2.1) is too long and sometimes overlapping with “application of biomimetic nanovaccines” paragraph 4. For instance, in the section 2.1.1 “Liposomes” the authors described a study (Moon et al) showing the development of a malaria vaccine based on NPs. I suggest to focus on the specific characteristics of different NPs in section 2 and describe their application in the specific section 4.

-on the contrary, the adjuvant section is not very exhaustive. Table 1 lists only few adjuvants among the many others evaluated so far. Since this review is not focused on the adjuvants, I suggest to include more papers as references, especially reviews on the subject and remove the table or specify that the table is a partial list of used adjuvants or under evaluation.

-reference 12 cited in the Introduction (row 57) is a review describing the mechanism of action of alum as an adjuvant, but the adverse effects are not mentioned. Please verify and add a proper reference. In addition, any adjuvant may induce side effects, not only alum, as mentioned here in the paper.

-Table 2 should be better edited/designed so that the corresponding descriptions are at the same level between the 2 columns.

-The authors described the advantages of nanovaccines, but they did not mention the possible limitations or disadvantages in the use of these platforms. It would be useful to include these aspects in section 3.

Author Response

(The authors gave the same response as above.)

Reviewer 3 Report

In the review, authors summarized the biomimetic nanoparticles, and their application in anti-bacterial therapy, anti-HIV therapy, anti-malarial therapy, anti-melittin therapy and anti-tumor immunity. The review is well organized and will be useful to readers in the related area. Therefore, it can be accepted after minor revision.

Authors should provide more discussion about the future direction. Authors should discuss the factors that influence the performance of biomimetic nanoparticles.

3.Except the cell membrane coating nanoparticles, exosomes are also widely used.

Section 3, the advantages of nanovaccines should be carefully summarized. The current discussion is too simple, and it is not enough to guide nanovaccine design by readers. There are other papers used biomimetic nanoparticles for cancer immunity (Biomaterials, 2019, 217, 119309; Advanced Materials, 2019, 31, 1901586), authors should refer them.

Author Response

(The authors gave the same response as above.)
